# Shelf Life and Safety of Vacuum Packed HPP-Treated Soaked Cod Fillets: Effects of Salt Content and Multilayer Plastic Film

**DOI:** 10.3390/foods12010179

**Published:** 2023-01-01

**Authors:** Gianluigi Ferri, Carlotta Lauteri, Mauro Scattolini, Alberto Vergara

**Affiliations:** 1Faculty of Veterinary Medicine, Post-Graduate Specialization School in Food Inspection “G. Tiecco”, University of Teramo, Strada Provinciale 18, 64100 Teramo, Italy; 2Food and Health Veterinarian Consulting, 64015 Nereto, Italy

**Keywords:** food microbiology, cod fish, high-pressure processing, oxygen transmission rate, food safety

## Abstract

High microbiological quality standards, food safety, and environmental sustainability represent crucial topics in food production chains. For this reason, fish industries, which import salted and seasoned fish products from supplier countries, i.e., Norway, Denmark, USA (Alaska State), etc., have tried to reduce the salt content of each carton during transportation (reducing carbon emissions and the weight of major quantities of transported fish). In the present study, 360 differently processed fish fillet samples, belonging to the species *Gadus macrocephalus* caught in FAO zone 67, were microbiologically and chemically screened. This study aimed to provide original data concerning the applicability of sustainable solutions investigating the combined effects of salt content reduction combined with new recyclable multilayer plastic film packaging (vacuum skin packaging with two different oxygen transmission rate values). The microbiological results showed no substantial changes comparing the two differently salted products, highlighting their high hygienic characteristics which were also observed in their chemical analysis. The shelf life evolutions (comparing the two different studied plastic films) highlighted that, after 35 days from HPP treatments, bacterial loads gained high values, over 6 log cfu/g. This study highlights that, compared to the currently used plastic films, the results of the new and sustainable multilayer plastic films show that they can provide safe food matrices in combination with HPP technologies. Therefore, this preliminary investigation brings closer attention to alternative and environmentally sustainable production systems with their designs based on the multidisciplinary approach of food production systems.

## 1. Introduction

Safe and ready-to-cook food matrices have been largely required by modern consumers [1]. For this reason, fish industries that produce salted cod or salt-cured fish have developed industrially rehydrated products (soaking process) which are classified as ready-to-cook foodstuffs [2]. In this way, producers have satisfied consumers’ requests and have avoided the organoleptic and microbiological substantial changes in the muscle tissue due to the wrongly managed rehydration process in the home. Indeed, the rehydration process is a crucial step in which salted products increase in water content to 70–85% and reduce salt to 2–3% providing a perfect microenvironment for bacterial multiplication [2,3,4]. The abovementioned products present a limited shelf life due to the high total bacterial count of more than 6 log10 cfu/g (stored at refrigerated temperatures of 1–4 °C for 6 days) with organoleptic alterations registered at days 7–10 from rehydration. This condition is justified by the increasing values of water parameter activities which achieve levels at which substantial microbial multiplication can be determined [3].

This issue has been solved by the combination of innovative technologies and new packaging materials that can microbiologically stabilize food matrices. High-pressure processing (HPP) has widely demonstrated the ability to reduce microbial loads by inducing lesions on the bacterial wall and other structures which are lethal for pathogens and commensal strains [5]. The HPP technology can be used on different matrices including fish and other seafood products, prolonging the shelf life and avoiding substantial sensorial property alterations [6]. This last aspect was demonstrated by Oliveira et al. [7] highlighting that the use of HPP improved the shelf life of rehydrated cod fish products. From a multidisciplinary perspective, material engineering has provided new plastic packing films that can be used in the food industry [8]. They are formed by multilayer plastic films presenting a high barrier for oxygen migration from the environment within the product. These barriers reduce and delay aerobic bacterial growth. This concept is expressed by a specific parameter named the oxygen transmission rate (OTR) and it has important repercussions on the shelf life of products. Generally, industries recommend plastic films for foodstuff packing with an OTR value ranging from 1 to 10 cm^3^/m^2^/24 h, measured at 22 °C and 75% relative humidity (RH) to produce safe products. This parameter is strategic for vacuum-skin-packed foodstuffs, especially for the fish products trade [9]. More in detail, focusing on an eco-sustainable perspective, the introduction of new plastic multilayers as valid substitutions has become of crucial importance in reducing environmental microfiber pollution [10].

The objective of this study was to provide a chemical and microbiological evaluation of an integrated fish food supply chain. This study aimed to compare the microbiology and chemistry of differently processed cod fish products in trying to provide sustainable alternatives to the salt content and sustainable multilayer plastic films as innovative packaging means for conservation, the application of HPP technology, etc. The same matrices were successively exposed to HPP technology and shelf life evaluations were made. All foodstuffs were screened to provide original data concerning a crucial environmental topic: *The Eco-sustainability of food matrices*.

## 2. Materials and Methods

All analyzed samples were collected from the same batch and supplier, and a flowchart was created which represents the product processing, starting from raw materials, represented by differently salted and seasoned cod fish, and arriving at the final products—soaked vacuum skin-packed, and HPP-treated specimens. The scientific design following the industrial flow is schematically illustrated in Figure 1.

### 2.1. Samples Collection

The present study involved fish fillet samples belonging to the species *Gadus macrocephalus*, caught in FAO zone 67 corresponding to the North-Eastern Pacific Ocean. After fishing, all animals were promptly beheaded and eviscerated to avoid parasitic migration from the intestinal environment to the muscle tissues (i.e., nematodes as *Anisakis* spp.) in accordance with the EU Reg. No. 1276/2011 which includes the *Gadidae* family in the groups of fish species in which the abovementioned procedures must be promptly performed by food operators. Frozen products were successively exported to Norway where fish producers processed them as salted and seasoned products (using marine salt). At the end of this step, Norwegian producers pack salted products with different salt contents per carton. In half of the batch, the supplier put 4 kg of NaCl as consuetudinary, and the other half of the batch was packed with 2 kg of NaCl. In each carton, an amount of 20–24 fillets were transported. Samples were imported by an Italian cod fish industry which was involved in this investigation. Sampling activities lasted six months [involving three different production periods (January, April, and July) upon product arrivals, with an average population of 100 kg/fish/transportation] and 360 differently processed fish samples starting from salted cod fish fillets characterized by an average weight of 400 ± 20 g/fillet (muscle tissue and skin), an average salt content of 16.5–17.4%, a pH of 6.4, and activity water (aw): 0.75. The screened population was considered statistically representative and in agreement with the industrial productive volumes which depend on seasonal catching in tons. These screened samples were obtained from splits of 1.2 ± 0.3 Kg (carton final weight 25 kg which includes fish and salt content), and the cutting step was performed by the industrial food operators. Following the above-reported flowchart (See Figure 1), salted and seasoned *G. macrocephalus* fillets were collected and introduced in the second analyzed technological process: the soaking step (6 days with refrigerated water). At the end of the soaking process (before HPP treatment), fillets were vacuum-skin-packed with two different plastic films characterized by two OTR values (OTR1 5.0 cm^3^/m^2^/24 h and OTR2 7.0 cm^3^/m^2^/24 h). The last industrially applied technology was the HPP treatment (setting: 600 MPa for 5 min). A further description forms part of the following paragraphs and sample types are summarized in Table 1.

The selected and tested production plastic films, that represented possible environmental and sustainable solutions, were selected for their material characteristics due to their reduction in polystyrene film layers (Cryovak^®^, Sealed Air, Milano, Italy).

The following sections include precise descriptions regarding the technological processes to which the products were exposed.

### 2.2. Samples Preparation

Fish fillets (400 ± 20 g/fillet) were soaked in a steel tank using refrigerated water at a set and continuously monitored temperature of + 2°C for 5–6 days. During rehydration, the tank water was systematically changed after specific time intervals of 6, 12, 18, and 24 h, and replaced with precooled water. This exchange was rapidly performed over a 5 min period using an automatic industrial pump system.

### 2.3. Packaging and HPP Treatment

Soaked fish fillet samples presented the following physicochemical characteristics: average weight of 480 g, 1.85% NaCl, pH 6.5, and aw 0.96. Samples were partially drained and successively vacuum skin-packed. Specimens were put in two different plastic packs characterized by two different OTR values. In detail, specimens were put in the pack named OTR1 (identified as *classic film*) with an OTR value of 5.0 cm^3^/m^2^/24 h (measured at 22 °C, 75% RH), and in the second one named OTR2 (identified as *new film*) with an OTR value of 7.0 cm^3^/m^2^/24 h (measured at 22 °C, 75% RH), as previously reported in Table 1. After rehydration and packing, the samples were introduced to the HPP machine, a QFP 2L-700 MPP produced by Avure Technologies^TM^—HPP, USA, and exposure parameters of 600 MPa for 5 min were set for their high bactericidal effect. These parameters have demonstrated the ability to extend product shelf life for over 40 days, as previously reported by Rode and Rotabaak [11]. After the HPP treatment, both differently packed (OTR1 and OTR2) samples were microbiologically analyzed and were involved in a shelf life evaluation. The quantitative and qualitative microbiological analysis started from day 0 [(t:0) immediately after HPP exposure] and continued on days t:15, t:30, and t:45. During these periods, all specimens were stored at a refrigerated temperature (+4 °C). In each sampling, specimens were collected directly from their respective production lines and were transported under refrigerated conditions to the laboratory; analyses were performed within 8 h after collection.

### 2.4. Chemical Analysis

Only raw material (salted and seasoned) samples were screened for heavy metal detection, i.e., cadmium (Cd), lead (Pb), and mercury (Hg) using a mass spectrometer (ICP-MS Instrument, Agilent) starting with homogenized samples; while, for all sample types, the pH (pH-meter, METTLER TOLEDO^©^, Milano, Italy), aw (AQUALAB 4TE METER FOOD^®^, Munich, Germany), and NaCl content through the usage of a salinometer (OPTIMARE, Precision Salinometer, Bremerhaven, Germany) were measured. Chemical analyses were performed following international standardized methods, as indicated in Table 2.

### 2.5. Microbiological Analysis

Aliquots of 25 g sectioned from muscle tissue were aseptically collected and introduced in stomacher bags (BagMixer^®^, Interscience, Puycapel, Cantal, France) adding 225 mL of maximum recovery diluent (MDR, Oxoid Ltd., UK). These samples were successively stomached for two minutes at room temperature to produce 10-fold dilutions. All incubated specimens were plated onto plate count agar (PCA, Thermo Scientific™ Oxoid Standard Plate Count Agar) and the obtained quantitative data were expressed in log cfu/g. The microbiological analysis also included the detection of pathogenic bacteria which are also reported in the EU Reg No. 2073/2005, and other bacteria more frequently identified in processed fish products. Selective culture media were performed following standardized methods, as reported in Table 3.

### 2.6. Statistical Analysis

Statistical procedures were performed using IBM^®^ SPSS^®^ Statistics [Version: 29.0.0.0 (241)]; analysis of dependent variable “TMC” and “TPC” values were respectively performed using the two-tailed paired *t*-test comparing two product types/analysis (“S4” vs. “R4”, “S2” vs. “R2”, “R4” vs. “R2”, and “S4” vs. “S2”). An assumption of normality was performed using Shapiro–Wilk test, and the alpha was 0.05. Concerning shelf life evaluations, the analysis of “TMC (TMC 0, TMC 15, TMC 30, and TMC 45)” and “TPC (TPC 0, TPC 15, TPC 30, and TPC 45)” values as dependent variables, obtained from R4 and R2 HPP-treated samples, were performed with Friedmann tests with TMC (TMC 0, TMC 15, TMC 30, and TMC 45) and TPC (TPC 0, TPC 15, TPC 30, and TPC 45) as within-subject factors. Post hoc comparisons were executed with the signed-rank Wilcoxon test and corrected with the Bonferroni procedure. Alpha was 0.05. Furthermore, data produced by the correlation between TMC and TPC (expressed as log-transformed data) values, obtained from differently packed (OTR1 and OTR2) HPP-treated products (coming from raw materials, salted fish transported with 4 kg salt/carton and 2 kg salt/carton, respectively), in the shelf life analysis (from t:0 to t:45). All raincloud plots were realized using JASP System (open-source software; version 0.16.4).

## 3. Results

### 3.1. Chemical Analysis

In all samples, heavy metals (Cd, Pb, and Hg) were not detected (resulting in less than 0.01 mg/kg). Sodium chloride content and aw presented different values from salted to rehydrated products, after the soaking process, as reported in Table 4.

Among the screened products, pH values were not statistically different if compared with the different salted products in each production step. Concerning the aw parameter, between S4 and S2, R4 and R2, HPP OTR1.4,2 and HPP OTR2-1,2, significant statistical differences were not observed. The NaCl content presented significant statistical differences comparing S4 and S2 with R4 and R2 products with a *p*-value (<0.0001). 

### 3.2. Microbiological Analysis: Salted and Soaked Products

In all salted and rehydrated samples and HPP-treated products, the obtained results have been compared with maximum values referring to the European Commission cut-offs and relative bibliographic limits established for specific bacterial strains, as illustrated in the following Table 5.

Bacterial pathogens such as *L. monocytogenes*, *Salmonella* spp., *V. cholerae,* and *V. parahaemolyticus* were not detected, and for the total coliform, *Enterobacteriaceae,* and *E. coli* counts were not found, presenting values under the respective cutoff values (less than 1 log cfu/g). 

The S4 samples showed a TMC average value of 2.86 ± 0.44 log cfu/g and a TPC average value of 2.75 ± 0.33 log cfu/g. S2 registered the following bacterial loads: TMC 2.80 ± 0.43 log cfu/g and TPC 2.52 ± 0.28 log cfu/g. The *t*-test conducted on the TMC S4 and S2 did not reveal a significant difference regarding the comparison of mesophilic counts [t(59) = 0.454; *p* = 0.651; Cohen’s d = 0.059] and the null hypothesis was accepted (TMC S4 = TMC S2); however, the difference between TPC S4 and TPC S2 values results were significant [t(59) = 4.133; *p* < 0.001; Cohen’s d = 0.534] (See Figure 2).

Coagulase-negative staphylococci amounts were 2.3 ± 0.15 log cfu/g in S4 products [25/60 (41.67%)] and 2.7 ± 0.21 log cfu/g in S2 products [28/60 (46.67%)]. *Pseudomonas* spp. count value was 1 log cfu/g in 5/60 S4 samples (8.33%), but it was not detected in S2 specimens (value less than 1 log cfu/g). In both sample types (S4 and S2), the lactic bacterial count results were under the method’s cutoff value. Coagulase-positive staphylococci enumeration also presented values less than 1 log cfu/g, but other staphylococci (coagulase-negative) were observed in both R4 [36/60 (60.0%)] and R2 [39/60 (65.0%)] products (2.60 ± 0.12 log cfu/g and 3.25 ± 0.23 log cfu/g, respectively).

Among soaked products, TMC R4 (3.51 ± 0.68 log cfu/g) and TMC R2 (3.50 ± 0.71 log cfu/g) were not significantly different [t(59) = 0.285; *p* = 0.777; Cohen’s d = 0.037 (mean ± SE: 3.51 ± 0.091)] (See Figure 2), but the psychrophilic strain counts, TPC R4 2.64 ± 0.48 log cfu/g and TPC R2 2.81 ± 0.39 log cfu/g were statistically different [t(59) = 2.065; *p* = 0.043; Cohen’s d = 0.267 (mean ± SE:2.752 ± 0.057)]. *Pseudomonas* spp. was also detected in R4 samples, originating from the same batch of positive S4 specimens, gaining a final amount of 2.1 log cfu/g. This last pattern was not described in the R2 products. Total lactic counts and coagulase-positive staphylococci enumeration screenings did not report positive results. The *t*-test conducted on the TMC S4 and R4 revealed a significant difference regarding the comparison of mesophilic counts [t(59) = 6.258; *p* < 0.001; Cohen’s d = 0.808 (mean ± SE: 3.19 ± 0.074)], and also TMC S2 and R2 presented significant differences [t(59) = 6.699; *p* < 0.001; Cohen’s d = 0.865 (mean ± SE: 3.16 ± 0.074)]. Concerning the TPC data, S4 and R4 showed significant differences [t(59) = 1.550; *p* < 0.001; Cohen’s d = 0.45 (mean ± SE: 2.764 ± 0.053)], and between S2 and R2, a similar result was also observed [t(59) = 4.924; *p* < 0.001; Cohen’s d = (mean ± SE: 2.521 ± 0.044)], as illustrated in the following Figure 3 and Figure 4.

### 3.3. HPP Products, OTR, and Shelf Life Evaluation

Among screened products, bacterial pathogens were not observed in the HPP-treated specimens and the same trend was also observed during the shelf life evaluation. Concerning TMC and TPC values observed from t:0 through to t:45 (Figure 5), the Friedmann test revealed significant differences between the different timing steps (shelf life), as reported in Table 6.

From the statistical evaluation of shelf life, significant differences between the timing steps (from t:0 to t:45) were also observed; representative data are illustrated in the following Table 6. Friedmann tests and relative post hoc procedures were used to reveal statistical differences (*p*-value < 0.05 characterized by high effect sizes r ≥ 0.50) in HPP-treated products that were vacuum packed with two different OTR coefficients. Table 7 and Table 8 report the observed data for TMC and TPC.

## 4. Discussion

Salted fish food matrices have represented safe and durable protein sources for human nutrition since ancient times [3]. These products are characterized by any nutritional aspects that provide a particular biochemical microenvironment that is inhospitable for many bacterial strains and other pathogens [4]. Nowadays, food-producing industries are strongly encouraged to produce sustainable foodstuffs by reducing their carbon footprints, resource misuse, and water consumption. This challenge has stimulated scientists and industrial partners to find common solutions [28]. Furthermore, in the salted and seasoned cod fish production chain, salt represents a waste product that is difficult to recycle, and for this reason, it must be cleared out. In accordance with a sustainable perspective, the possibility of reducing the salt content in each fish transportation carton, transportation that can last for different months, could permit the introduction of more fish in each package (more fish tons), and consequently a decrease in salt consumption and a reduction in carbon emissions. For this reason, in this research, a preliminary qualitative and quantitative microbiological evaluation was undertaken to compare bacteriological and chemical evidence obtained from salted and seasoned fillet samples traded and transported with different salt contents in each carton. Secondly, the application of innovative physical food technologies (i.e., HPP as “*no-thermal sterilization*”) coupled with a comparative analysis between two different plastic films (as previously defined: OTR1—classic film and OTR2—new film) have provided interesting data to the real-world applicability of innovative and ecological multilayer plastic films in the fish food industry, also guaranteeing safe matrices.

Starting with chemical parameters, the results were in line with previous studies [4,11,28]. From a microbiological analysis, bacterial pathogens such as *L. monocytogenes*, *E. coli*, *Salmonella* spp., and *Vibrio* spp. were not detected, also in line with previous research investigations conducted on salted fish matrices [11,29,30]. Their absence could firstly be related to the inhospitable salted microenvironment, and secondly to the proper hygiene (GHP) and manufacturing practices (GMP) applied during food processing, in agreement with the international standard checklists used by the sample supplier [29]. Between the TMC values of S4 and S2 products, there were no significant differences, as explained in the results section, accepting the null hypothesis which affirms that TMC S4 and TMC S2 were not statistically different. This condition suggested similar quantitative microbiological loads for the two screened and differently salted food matrices (containing different salt/carton 4.0 and 2.0 kg NaCl/carton). These preliminary findings and the absence of bacterial pathogens exclude potential risks for consumers and demonstrate that products transported in cartons with lower salt contents can last for 6 months without microbiologically influencing product characteristics. These data supported the evidence that a consistent salt reduction during transportation did not significantly influence products’ sanitary peculiarities. The soaking process which followed the specimen salting sections showed significant differences in both TMC and TPC values when comparing salted matrices with respective rehydrated matrices. These findings were explained by the water re-uptake levels during the soaking process, as demonstrated by recent studies [31,32], and in agreement with data observed and reported in Table 7 and Table 8 (Results section). Focusing on the physical parameters, the average activity water (aw) values increased from 0.75 to a final aw of 0.96, in agreement with the scientific observations reported by Rodrigues et al. [33]. In these microenvironmental conditions, both Gram-positive and Gram-negative microorganisms find favorable conditions for bacterial multiplication [3]. For these reasons, the shelf life of soaked products is included in a range of 3–5 days, if stored at refrigerated conditions [34].

The introduction of HPP technologies as a microbiological stabilization method (reducing several pathogens and commensal multiplications) provides innovative and sustainable products which are conservable for many days (mean of 35 days after HPP treatment) [35]. Indeed, in this scientific investigation, the HPP exposures determined the TMC and TPC values which decreased from 2.6 log10 cfu/g (TMC) from soaked fillets to the HPP-treated (t:0) samples with final medium values of TMC of 1.3 log10 cfu/g and TPC of 1.5 log10 cfu/g in all products. These results highlighted the well-known efficacy of this technology and the obtained data were in line with that which was previously demonstrated by Rode and Rotabakk [10]. This agreement was justified by using the same HPP machine exposure setting of 600 MPa for 5 min and a process that has been widely identified as the gold standard (defined as “*novel non-thermal processing technology*”) for food matrices, as demonstrated by many scientific papers [36,37,38]. The HPP energy costs were 0.16 KWh/kg of fish fillets (higher if compared with thermal costs, i.e., 0.04 KWh/kg fish fillets for the microbiological stabilization) and in agreement with data presented by Aganovic et al. [39] that observed 0.20 KWh/l using the same HPP setting as used in the present study (600 MPa for 5 min). However, this first energetic gap has been widely filled by the solid implementation of solar systems which have been involved in the independency of many industrial realities, as widely demonstrated by Gulzar et al. [40].

In the Results section, Table 7 and Table 8 provide a complete illustration of this dynamic process in which TMC and TPC values changed from day t:0 through to t:45, highlighting significant differences between the different time steps. The observed trends were quite different when compared to the Rode and Rotabakk [11] results. The results of the first part of this shelf life evaluation were in line with the previously mentioned authors, until day t:30; after this time limit, the TPC increases presented statistical differences greater than TMC, gaining final loads of 6.0 log cfu/g, 1.5 log cfu/g more than Rode and Rotabaak [11].

In the comparison between HPP-soaked products packed with two different plastic films characterized by specific OTR values (OTR1: 5.0 cm^3^/m^2^/24 h and OTR2 7.0 cm^3^/m^2^/24 h), statistical analysis revealed significant differences along with the shelf life evaluation, as represented in Table 7 and Table 8, also highlighting the notable increases in TPC (especially after t:15) when compared to TMC. This finding was justified by the conservation of the food matrices at refrigerated conditions which mainly determine psychrophilic bacteria proliferation [41]. After t:30, as described above, the strong multiplication was related to the different oxygen migration levels through the plastic films, and this last aspect had repercussions on bacterial loads. Comparing OTR1 and OTR2 packed products, the bacterial increases presented significant differences between the two studied films. Indeed, after 45 days (t:45), TMC and TPC final levels were 6.0 log10 cfu/g for OTR-1 and 7.0 log10 cfu/g for OTR-2. These results were different from the values reported by Rode and Rotabaak [10] who, at day 49, observed a bacterial load below 5 log10 cfu/g. This gap was justified by the different OTR plastic films used in the screened vacuum-skin-packed HPP-soaked products. Different from the previous study, the obtained results were lower than those reported at stage t:0. In both cases, the high impact and efficacy of the HPP technology on shelf life prolongation and its damaging actions on bacterial walls and ultra-structures were confirmed. In this study, t:30 was identified as a potential cutoff to be considered as a possible expiration date due to the evidence that suggested that after this time, when following the described processing conditions, bacterial proliferation has determined organoleptic alterations. Sustainable plastic multilayer films (different OTR values with high recyclability scores) for fish matrices combined with consolidated stabilizing food technologies have provided the real applicability of innovative measures at the industrial level.

A final consideration focuses on qualitative microbiological aspects. *Staphylococcus* spp. (excluding coagulase-positive strains) was the most representative microbial population both in the salted and soaked fish products, agreeing with previous studies [33,42], but were strongly reduced in HPP-treated (t:0) specimens, with final counts less than 1 log cfu/g. These results agree with the scientific hypothesis formulated by Rode and Rotabkk [11] and Dhar et al. [30] which demonstrated the high efficacy of hydrostatic pressures on Gram-positive strains. *Pseudomonas* spp. and lactic bacterial counts presented low values which were further reduced by the HPP actions, resulting in final levels of less than 1 log10 cfu/g (t:0). Their detection is considered a normal microbiological condition because these strains are considered spoilage strains, as identified from soaked cod fish products [33].

Finally, from this study, in applying an interdisciplinary approach conjugating food microbiology to innovative food processing technologies (materials and mechanical engineering), we wanted to provide a scientific representation of an integrated fish processing supply chain. This study has served to highlight, at the food industry level, the recent evolution in innovative production systems and their respective environmental sustainability (salt content reductions for each transporting carton resulting in major productive volume reductions, less salt wasting, and lower carbon emissions) compared to the traditional salted cod fish trades, and the practical applicability of new multilayer films with high recyclability, as a means of reducing environmental pollution. The obtained findings supported the consideration that collaboration between science and the food industries represents the first crucial step in pragmatically applying the so-called one-health approach. This aspect can lead to an initial but substantial decrease in chain carbon releases which have notable repercussions on the calculation of sustainability reports, as indicated in the ISO 14016:2020, which represents an important document that industries should have.

In conclusion, it is possible to affirm that a wide scientific perspective is mandatory in assessing the “*contaminations*” of different disciplines which pose a threat to human, animal, and environmental health.

## 5. Conclusions

The integrated fish food production chain was studied. In revising the salt content of supplied salted raw materials, interesting microbiological data were presented ensuring safe products for the consumer. Furthermore, among the production processes, high hygienic standards combined with efficient technologies, i.e., HPP, and new packaging materials, can be valid alternatives to the classic films to give substantial solutions to the growing environmental necessity of sustainability. In this way, it is possible to provide products characterized by an extended conservation period. The employment of material engineering and new food technologies, starting with an increased knowledge of the microenvironment of food matrices, is fundamental for the production of safe perishable food products.

## Figures and Tables

**Figure 1 foods-12-00179-f001:**
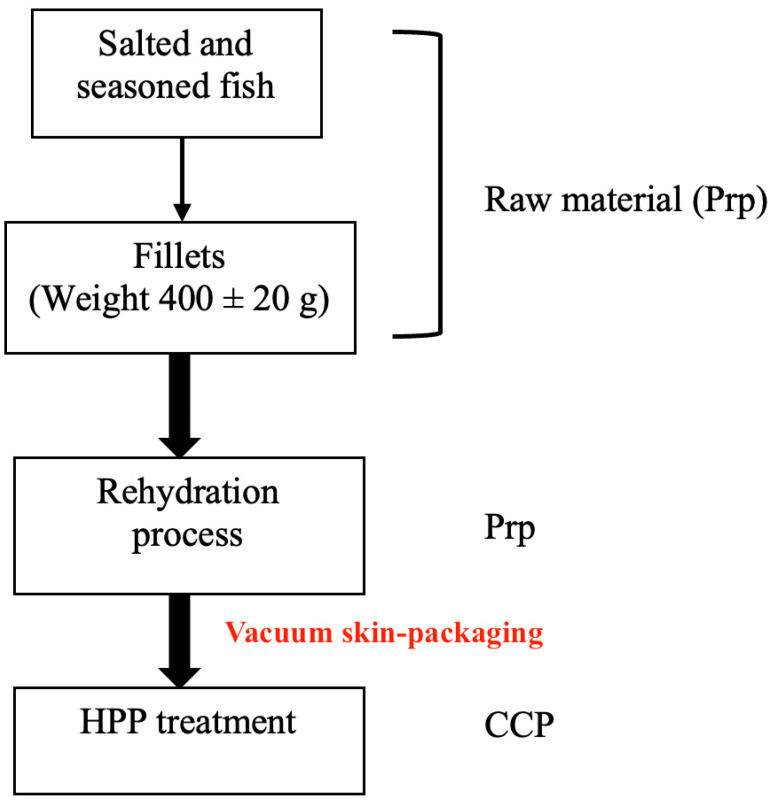
Article flowchart. Prp: prerequisite; CCP: critical control point.

**Figure 2 foods-12-00179-f002:**
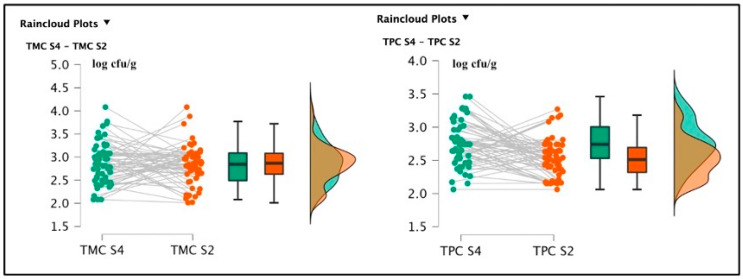
Raincloud plots representing the TMC and TPC of differently salted screened samples. For each graphic, the vertical axes report the calculated bacterial loads, and the horizontal axes indicate the screened TMC (on the **left**) or TPC (on the **right**) from the analyzed food matrices (S4–S2).

**Figure 3 foods-12-00179-f003:**
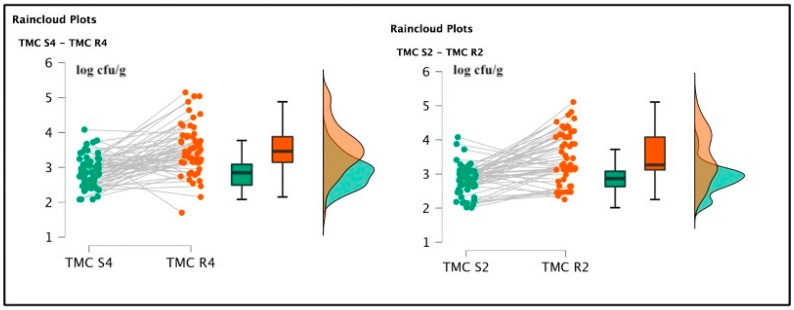
Raincloud plots representing TMC value distributions from differently salted to rehydrated products. For each graphic, the vertical axes report the calculated bacterial loads, and the horizontal axes indicate the screened TMC values from S4–R4 (on the **left**) and S2–R2 (on the **right**) analyzed food matrices.

**Figure 4 foods-12-00179-f004:**
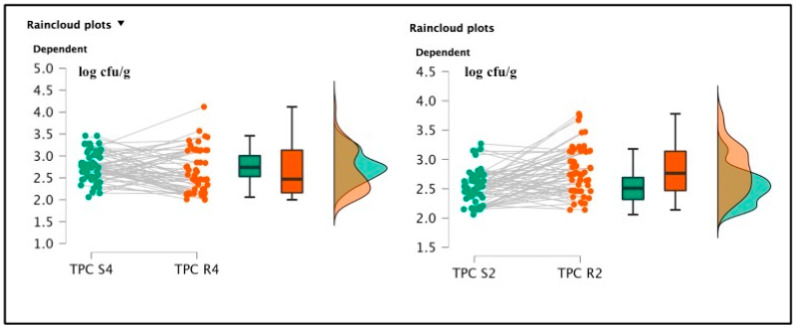
Raincloud plots representing TPC value distributions from differently salted to rehydrated products. For each graphic, the vertical axes report the calculated bacterial loads, and the horizontal axes indicate the screened TPC values from S4–R4 (on the **left**) and S2–R2 (on the **right**) analyzed food matrices.

**Figure 5 foods-12-00179-f005:**
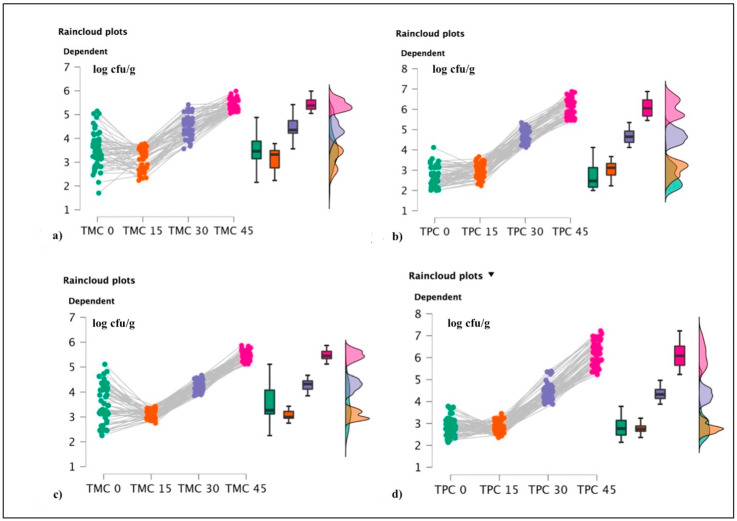
Raincloud plots representing. For each graphic, the vertical axis reports the calculated bacterial loads, and the horizontal axis indicates TMC and TPC values during the shelf life evaluations. More in detail, (**a**) TMC values for OTR1; (**b**) TPC for OTR1; (**c**) TMC for OTR2; and (**d**) TPC for OTR2.

**Table 1 foods-12-00179-t001:** Samples included in the present investigation.

Samples	Fish Species	Processed Products
360 fish products	*Gadus macrocephalus*	60 S4
		60 S2
		60 R4
		60 R2
		30 HPP OTR1-4
		30 HPP OTR1-2
		30 HPP OTR2-4
		30 HPP OTR2-2

**S4:** salted samples (4 kg NaCl/carton); **S2:** salted samples (2 kg NaCl/carton); **R4:** rehydrated samples (4 kg NaCl/carton); **R2:** rehydrated samples (2 kg NaCl/carton). **HPP OTR1-4:** treated product OTR1 packed with salted products transported with 4 kg NaCl/carton. **HPP OTR1-2:** treated product OTR1 packed with salted products transported with 2 kg NaCl/carton. **HPP OTR2-4:** treated product OTR2 packed with salted products transported with 4 kg NaCl/carton. **HPP OTR2-2:** treated product OTR2 packed with salted products transported with 2 kg NaCl/carton.

**Table 2 foods-12-00179-t002:** Chemical parameters calculated in the present study in accordance with specific international standards.

Chemical Parameters	International Standard	Description
Cd, Pb, and Hg (mg/Kg)	ISO 15763:2010	Foodstuffs—Determination of trace elements—Determination of arsenic, cadmium, mercury, and lead in foodstuffs by inductively coupled plasma mass spectrometry (ICP-MS) after pressure digestion [12].
	ISO 13805:2014	Foodstuffs—Determination of trace elements—Pressure digestion [13].
pH	ISO 2917:1999	Measurement of pH—Reference method [14].
aw	ISO 21807:2004	Microbiology of food and animal foodstuffs—Determination of water activity [15].
NaCl (percentage)	AOAC 937.09	Salt content (chlorine as sodium chloride) in seafood [16].

**Table 3 foods-12-00179-t003:** Microbiological screenings performed in this study in accordance with specific UNI EN ISO.

Bacterial Strain	Standards	Description
Total Mesophilic Count (TMC)	ISO 4833:20	Microbiology of the food chain—Horizontal method for the enumeration of microorganisms—Part 1: Colony count at 30 °C by the pour plate technique [17].
Total Psychrophilic Count (TPC)	ISO 17410:2019	Microbiology of the food chain—Horizontal method for the enumeration of psychrotrophic microorganisms [18].
*Listeria monocytogenes*	ISO 11290-1:2017 ISO 11290-2:2017	Horizontal method for the detection and enumeration of Listeria monocytogenes and Listeria spp.—Part 1: Detection method; Part 2: Enumeration method [19].
*Salmonella* spp.	ISO 6579-1:2020 ISO 6579-2:2020	Horizontal method for the detection, enumeration, and serotyping of Salmonella.—Part 1: Detection method; Part 2: Enumeration method [20].
Coagulase-positive staphylococci enumeration	ISO 6888-1:2018	Horizontal method for the enumeration of coagulase-positive staphylococci (Staphylococcus aureus and other species)—Part 1: Method using Baird–Parker agar medium [21].
*Pseudomonas* spp.	ISO 13720:2010	Enumeration of presumptive *Pseudomonas* spp. [22].
*Escherichia coli* (positive beta-glucuronidase)	ISO 16649-2:2001	Horizontal method for the enumeration of beta-glucuronidase-positive Escherichia coli—Part 2: Colony count technique at 44 degrees C using 5 bromo-4-chloro-3-indolyl beta-D-glucuronide [23].
Total Coliform count	ISO 4832:2006	Horizontal method for the enumeration of coliforms [24].
*Enterobacteriaceae* count	ISO 21528-2:2017	Horizontal method for the detection and enumeration of *Enterobacteriaceae*—Part 2: Colony count technique [25].
*Vibrio cholerae* and *V. parahaemolyticus* count	ISO 21872-1:2017	Horizontal method for the determination of Vibrio spp.—Part 1: Detection of potentially enteropathogenic *Vibrio parahaemolyticus*, *Vibrio cholerae,* and *Vibrio vulnificus* [26].
Total lactic bacterial count	ISO 15214:1998	Horizontal method for the enumeration of mesophilic lactic acid bacteria—Colony count technique at 30 degrees C [27].

**Table 4 foods-12-00179-t004:** pH, aw, and sodium chloride detection from differently processed cod products (average and standard deviation).

Products	pH	aw	NaCl (%)
S4	6.4 ± 0.1	0.75 ± 0.01	17.65 ± 1.13%
S2	6.2 ± 0.1	0.76 ± 0.02	17.14 ± 0.72%
R4	6.7 ± 0.2	0.96 ± 0.01	1.94 ± 0.45%
R2	6.6 ± 0.1	0.96 ± 0.01	1.74 ± 0.37%
HPP OTR1-4	6.5 ± 0.1	0.97 ± 0.02	1.99 ± 0.46%
HPP OTR1-2	6.5 ± 0.1	0.96 ± 0.01	1.85 ± 0.42%
HPP OTR2-4	6.4 ± 0.2	0.96 ± 0.01	1.82 ± 0.35%
HPP OTR2-2	6.6 ± 0.1	0.96 ± 0.01	1.90 ± 0.44%

**Table 5 foods-12-00179-t005:** Maximum values and relative limits of microbiological factors obtained from this study.

Microbiological Factors	Obtained Results	Maximum Values
*Listeria monocytiogenes*	Absence	Absence in 25 g/products (EU Reg. No. 2073/2005)
*Salmonella* spp.	Absence	Absence in 25 g/products (EU Reg. No. 2073/2005)
Coagulase-positive staphylococci enumeration	Absence	1 log cfu/g [21]
*Pseudomonas* spp.		
*Escherichia coli*	Absence	Absence in 25 g/products (EU Reg. No. 2073/2005)
Total colifomr count	Absence	1 log cfu/g [24]
*Enterobatteriacea* count	Absence	1 log cfu/g [25]
*Vibrio cholarae* and *V. parahaemolyticus* counts	Absence	<1 log cfu/g [26]
Total lactic bacterial count	<1 log cfu/g	<1 log cfu/g [27]

**Table 6 foods-12-00179-t006:** Nonparametric statistical (Friedmann and Wilcoxon) results for shelf life evaluation.

**Friedmann Test**	**Sample Types**		**χ^2^**		**df**	**Asymp. Sig.**	
	**TMC 4 (t:0–t:45)**		153.183		3	<0.001	
	**TPC 4 (t:0–t:45)**		168.480		3	<0.001	
	**TMC 2 (t:0–t:45)**		148.820		3	<0.001	
	**TPC 2 (t:0–t:45)**		162.020		3	<0.001	
**Wilcoxon signed-rank tests**	**Post hoc comparisons**	**TMC4 t:0–15**	**TMC4 t:0–30**	**TMC4 t:0–45**	**TMC4 t:15–30**	**TMC4 t:15–45**	**TMC4 t:30–45**
	**Z**	−3.242	−6.088	−6.680	−6.736	−6.681	−6.680
	**Asymp. Sig. (2-tailed)**	*p* < 0.001	*p* < 0.001	*p* < 0.001	*p* < 0.001	*p* < 0.001	*p* < 0.001
	**r**	0.55	0.78	0.86	0.87	0.86	0.86
	**Post hoc comparisons**	**TPC4 t:0–15**	**TPC4 t:0–30**	**TPC4 t:0–45**	**TPC4 t:15–30**	**TPC4 t:15–45**	**TPC4 t:30–45**
	**Z**	−4.814	−6.736	−6.736	−6.737	−6.736	−6.736
	**Asymp. Sig. (2-tailed)**	*p* < 0.001	*p* < 0.001	*p* < 0.001	*p* < 0.001	*p* < 0.001	*p* < 0.001
	**r**	0.62	0.87	0.87	0.87	0.87	0.87
	**Post hoc comparisons**	**TMC2 t:0–15**	**TMC2 t:0–30**	**TMC2 t:0–45**	**TMC2 t:15–30**	**TMC2 t:15–45**	**TMC2 t:30–45**
	**Z**	−3.534	−5.694	−6.736	−6.738	−6.736	−6.736
	**Asymp. Sig. (2-tailed)**	*p* < 0.001	*p* < 0.001	*p* < 0.001	*p* < 0.001	*p* < 0.001	*p* < 0.001
	**r**	0.46	0.73	0.87	0.87	0.87	0.87
	**Post hoc comparisons**	**TPC2 t:0–15**	**TPC2 t:0–30**	**TPC2 t:0–45**	**TPC2 t:15–30**	**TPC2 t:15–45**	**TPC2 t:30–45**
	**Z**	−0.280	−6.736	−6.736	−6.737	−6.736	−6.736
	**Asymp. Sig. (2-tailed)**	*p* < 0.001	*p* < 0.001	*p* < 0.001	*p* < 0.001	*p* < 0.001	*p* < 0.001
	**r**	0.04	0.87	0.87	0.87	0.87	0.87

**χ^2^:** chi-squared; **df:** degrees of freedom; **Asymp. sig.:** asymptotic significance; **Z:** statistic value; **r:** statistical effect size.

**Table 7 foods-12-00179-t007:** Statistical results and respective post hoc inferences concerning TMC values and OTR1–2 packaging.

**Friedmann Test**	**Sample types**		**Time**	**χ^2^**	**df**	**Asymp. Sig.**
	**OTR1–4; OTR1–2; OTR2–4; OTR2–2**		**t:0**	25.456	3	<0.001
			**t:15**	46.145	3	<0.001
			**t:30**	41.010	3	<0.001
			**t:45**	10.030	3	0.018
**Post hoc evaluations**	**TMC t:0 OTR1–4 vs. OTR1–2**	**TMC t:0 OTR1–2 vs. OTR2–4**	**TMC t:0 OTR2–4 vs. OTR2–2**	**TMC t:0 OTR1–4 vs. OTR2–4**	**TMC t:0 OTR1–4 vs. OTR2–2**	**TMC t:0 OTR1–2 vs. OTR2–2**
**z**	−0.714	−1.938	−1.288	−2.996	−4.098	−3.438
**Asymp. sig.**	0.475	0.053	0.198	0.003	<0.001	<0.001
**r**	0.08	0.35	0.23	0.55	0.75	0.63
**Post hoc evaluations**	**TMC t:15 OTR1–4 vs. OTR1–2**	**TMC t:015 OTR1–2 vs. OTR2–4**	**TMC t:15 OTR2–4 vs. OTR2–2**	**TMC t:15 OTR1–4 vs. OTR2–4**	**TMC t:15 OTR1–4 vs. OTR2–2**	**TMC t:15 OTR1–2 vs. OTR2–2**
**z**	−2.757	−4.783	−4.618	−4.206	−0.951	−1.925
**Asymp. sig.**	0.006	<0.001	<0.001	<0.001	0.341	0.054
**r**	0.50	0.87	0.84	0.77	0.17	0.35
**Post hoc evaluations**	**TMC t:30 OTR1–4 vs. OTR1–2**	**TMC t:30 OTR1–2 vs. OTR2–4**	**TMC t:30 OTR2–4 vs. OTR2–2**	**TMC t:30 OTR1–4 vs. OTR2–4**	**TMC t:30 OTR1–4 vs. OTR2–2**	**TMC t:30 OTR1–2 vs. OTR2–2**
**z**	−1.341	−4.546	−4.372	−4.659	−3.159	−2.055
**Asymp. sig.**	0.180	<0.001	<0.001	<0.001	0.002	0.040
**r**	0.24	0.83	0.79	0.85	0.57	0.37
**Post hoc evaluations**	**TMC t:45 OTR1–4 vs. OTR1–2**	**TMC t:45 OTR1–2 vs. OTR2–4**	**TMC t:45 OTR2–4 vs. OTR2–2**	**TMC t:45 OTR1–4 vs. OTR2–4**	**TMC t:45 OTR1–4 vs. OTR2–2**	**TMC t:45 OTR1–2 vs. OTR2–2**
**z**	−1.990	−1.008	−0.076	−2.428	−2.613	−1.352
**Asymp. sig.**	0.047	0.313	0.940	0.015	0.009	0.176
**r**	0.36	0.18	0.01	0.44	0.47	0.25

**χ^2^:** chi-squared; **df:** degrees of freedom; **Asymp. sig.:** asymptotic significance; **Z:** statistic value; **r:** statistical effect size.

**Table 8 foods-12-00179-t008:** Statistical results and respective post hoc inferences concerning TPC values and OTR1–2 packaging.

**Friedmann Test**	**Sample Types**		**Time**	**χ^2^**	**df**	**Asymp. Sig.**
	**OTR1–4; OTR1–2; OTR2–4; OTR2–2**		**t:0**	21.949	3	<0.001
			**t:15**	31.455	3	<0.001
			**t:30**	62.185	3	<0.001
			**t:45**	72.564	3	<0.001
**Post hoc evaluations**	**TPC t:0 OTR1–4 vs. OTR1–2**	**TPC t:0 OTR1–2 vs. OTR2–4**	**TPC t:0 OTR2–4 vs. OTR2–2**	**TPC t:0 OTR1–4 vs. OTR2–4**	**TPC t:0 OTR1–4 vs. OTR2–2**	**TPC t:0 OTR1–2 vs. OTR2–2**
**z**	−0.854	−3.340	−0.444	−2.680	−3.010	−3.436
**Asymp. sig.**	0.393	<0.001	0.657	0.007	0.003	<0.001
**r**	0.15	0.61	0.08	0.49	0.55	0.63
**Post hoc evaluations**	**TPC t:15 OTR1–4 vs. OTR1–2**	**TPC t:015 OTR1–2 vs. OTR2–4**	**TPC t:15 OTR2–4 vs. OTR2–2**	**TPC t:15 OTR1–4 vs. OTR2–4**	**TPC t:15 OTR1–4 vs. OTR2–2**	**TPC t:15 OTR1–2 vs. OTR2–2**
**z**	−0.465	−4.314	−4.680	−4.105	−0.802	−0.865
**Asymp. sig.**	0.642	<0.001	<0.001	<0.001	0.422	0.387
**r**	0.08	0.78	0.85	0.75	0.15	0.15
**Post hoc evaluations**	**TPC t:30 OTR1–4 vs. OTR1–2**	**TPC t:30 OTR1–2 vs. OTR2–4**	**TPC t:30 OTR2–4 vs. OTR2–2**	**TPC t:30 OTR1–4 vs. OTR2–4**	**TPC t:30 OTR1–4 vs. OTR2–2**	**TPC t:30 OTR1–2 vs. OTR2–2**
**z**	−4.599	−4.762	−2.561	−4.536	−2.790	−4.705
**Asymp. sig.**	<0.001	<0.001	0.010	<0.001	0.005	<0.001
**r**	0.84	0.87	0.47	0.83	0.51	0.86
**Post hoc evaluations**	**TPC t:45 OTR1–4 vs. OTR1–2**	**TPC t:45 OTR1–2 vs. OTR2–4**	**TPC t:45 OTR2–4 vs. OTR2–2**	**TPC t:45 OTR1–4 vs. OTR2–4**	**TPC t:45 OTR1–4 vs. OTR2–2**	**TPC t:45 OTR1–2 vs. OTR2–2**
**z**	−1.094	−4.783	−0.720	−4.783	−4.783	−4.783
**Asymp. sig.**	0.274	<0.001	0.472	<0.001	<0.001	<0.001
**r**	0.20	0.87	0.13	0.87	0.87	0.87

**χ^2^:** chi-squared; **df:** degrees of freedom; **Asymp. sig.:** asymptotic significance; **Z:** statistic value; **r:** statistical effect size.

## Data Availability

Data is contained within the article.

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
