# Peer review of "Shelf Life and Safety of Vacuum Packed HPP-Treated Soaked Cod Fillets: Effects of Salt Content and Multilayer Plastic Film"

_foods, 2023, doi:10.3390/foods12010179_

Round 1

Reviewer 1 Report

The manuscript entitled "Shelf-life and safety of vacuum packed HPP-treated soaked codfish fillets: Effects of salt contents and multilayer plastic film" by Ferri  G et al. compared microbiological and chemical parameters of differently processed fish fillets samples. This study's results could be useful food industry for the shelf-life and safety of vacuum-packed HPP-treated ready-to-cook foodstuffs. However, the manuscript needs some revisions as follows;

Generally, The English content of the manuscript is understandable, however, there a lot of  grammatical and technical errors. The manuscript requires a thorough proofread by a native person whose first language is English. Also my minor revisions should be addressed:

Abstract

Abastract is unclear. The aim of this study should be written more clearly in the abstract section. Also, the authors should be given more detail about chemical and antimicrobial results.

Indtroduction

Line 30 should be rewritten

Line 43. "expressed as log cfu/g" should be removed

Line 52. "multiplication" should be growth

Line 55: “ranging  between” should be changed as  ranging from … to..

Line 56, 131, 133, line 359 ....:  cm3/m2/  3 and 2 should be written as a superscript

Line 61-72. this section should be rewritten to be short and  clear and lines 63 to 70 are suitable for the material and method section.

Materials and Methods

Line 74: "screened" should be "analyzed"

Line 90: “exported in Norway” should be “exported to Norway”

the sub-headings: 2.1 and 2.2 are the same. These two sub-headings should be given a one sub-headings, or 2.2.  sample collection should be changed as sample preparation

Line 121: “set on” should be “set at”

Line 137. “impact1 should be “effect”

Lines 156-162 is not necessary, I suggest that I suggest that the total mesophilic count (TMC) 156 and the total psychrophilic count (TPC) and related methods should be placed in table 3 and this sentence in lines 153-162 should be removed.

Results

The results were well presented. The authors adequately assessed the results with the statistical method.

Authors should better explain the content of the image in Figures 2 to 5

Author Response

Dear Reviewer 1,

We are grateful for the provided suggestions. Following Reviewer’s indications, we think that our manuscript has improved its scientific quality. All suggested revisions were performed. In the following paragraphs we reported all changes.

Reviewer 1 suggestions:

Reviewer 1

Abstract is unclear. The aim of this study should be written more clearly in the abstract section. Also, the authors should be given more detail about chemical and antimicrobial results.

Authors (lines 16-23): The Abstract Section has been revised.

Introduction

Line 30 should be rewritten.

Authors (lines 32-33): This sentence has been rewritten.

Line 43. "expressed as log cfu/g" should be removed.

Authors (line 77): "expressed as log cfu/g" has been removed.

Line 52. "multiplication" should be growth.

Authors (line 86): "multiplication" has been replaced with “growth”.

Line 55: “ranging  between” should be changed as  ranging from … to..

Authors (line 89): “ranging  between” has been replaced with ranging from … to..

Line 56, 131, 133, line 359 ....:  cm3/m2/  3 and 2 should be written as a superscript.

Authors (lines 89, 221, 223, 515): this part has been changed.

Line 61-72. this section should be rewritten to be short and  clear and lines 63 to 70 are suitable for the material and method section.

Authors (lines 148-151): This part has been moved from Introduction to Material and Methods Section.

Line 61-72. this section should be rewritten to be short and  clear and lines 63 to 70 are suitable for the material and method section.

Authors (lines 95-103): This part has been modified.

Materials and Methods

Line 74: "screened" should be "analyzed".

Authors (line 105): "screened" has been replaced with  "analyzed".

Line 90: “exported in Norway” should be “exported to Norway”

Authors (line 152): “exported in Norway” has been replaced with “exported to Norway”.

The sub-headings: 2.1 and 2.2 are the same. These two sub-headings should be given a one sub-headings, or 2.2.  sample collection should be changed as sample preparation.

Authors (line 226): “2.2” has been changed as “2.2 samples preparation”.

Line 121: “set on” should be “set at”.

Authors (line 227): “set on” has been replaced with “set at”.

Line 137. “impact1 should be “effect”.

Authors (line 243): “impact” has been replaced with “effect”.

Lines 156-162 is not necessary, I suggest that I suggest that the total mesophilic count (TMC) 156 and the total psychrophilic count (TPC) and related methods should be placed in table 3 and this sentence in lines 153-162 should be removed.

Authors (lines 279-282): This part has been modified.

Results

Reviewer 1: Authors should better explain the content of the image in Figures 2 to 5

Authors: Figures (from Fig. 2 to Fig. 5) have been modified.

We confirm that neither the manuscript nor any parts of its content are currently under consideration or published in another journal.

All authors have approved the manuscript and agree with its submission to the FOODS Journal.

We appreciate the possibility to publish our paper and believe that our revised manuscript will be of interest to You and to the readers of Your journal.

Thank You for Your time and attention.

Best regards,

Gianluigi Ferri

Doctor in Veterinary Medicine (D.V.M.)

Ph.D. Student in Food Inspection

Faculty of Veterinary Medicine; University of Teramo, Italy.

Reviewer 2 Report

The article "" has been revised. finding a high-quality manuscript with a solid statistical base that supports the research work.

As general comments, it should be noted that in the methodological part it is necessary to describe the equipment used to measure certain physical and chemical parameters.

Additionally, for the results of the microbiological analyzes, it may be helpful to have a table and compare what is obtained with the maximum values allowed for each determination.

Finally I want to highlight that the discussion is well achieved and the general merit is high.

Minor comments were described within the attached document.

Author Response

Dear Reviewer 2, 

We are grateful for the provided suggestions. Following Reviewer’s indications, we think that our manuscript has improved its scientific quality. All suggested revisions were performed.

In the following paragraphs we reported all changes.

Reviewer 2 suggestions:

Lines 95-97: How was determined the 360 number for sample collection?Authors (lines 160-162): Following reviewer’s suggestions (.pdf file), all required information has been added. According to the transported fish ton amounts, n. 360 products were considered statistically representative concerning the industrial productive volumes.

Line 101: it is not explained how the experimental design was determined. Authors (lines 166-176): Following reviewer’s suggestions, this part has been improved.

Line 178: why Shapiro-Wilk? justify why.. Because this statical tests is for a size sample of n=50, maximum.                                                                                Authors: Shapiro-Wilk test has been performed for the assessment of normality as preliminary statistical evaluation to check the normality factor and value distribution before calculating he two-tailed paired T-test. Its application has been conducted in agreement with Ghasemi, A., & Zahediasl, S. (2012). Normality tests for statistical analysis: a guide for non-statisticians. International journal of endocrinology and metabolism, 10(2), 486–489. https://doi.org/10.5812/ijem.3505.).

Reviewer 2: Talking about sustainable processes. What is the energy cost of using HPP per kilo of cod processed vs a standard heat treatment.            Authors (lines 570-576): Following reviewer’s suggestions, these sentences have been implemented.

Reviewer 2: As general comments, it should be noted that in the methodological part it is necessary to describe the equipment used to measure certain physical and chemical parameters.                                                                                  Authors (lines 253-258): Following reviewer’s suggestions, this part has been improved.

Reviewer 2: Additionally, for the results of the microbiological analyzes, it may be helpful to have a table and compare what is obtained with the maximum values allowed for each determination.                                                                          Authors (line 352): Following reviewer’s suggestion, Table 5 has been introduced in the Results Section.

Reviewer 3 Report

Respected authors,

The manuscript is well written, clearly presented and interesing to read with results that are applicable in practice. 

How do you explain that samples with higher salt content have higher TMC and TPC (The S4 samples showed a TMC average value of 2.86±0.44 log cfu/g and TPC  2.75±0.33 log cfu/g, respectively. S2 registered the following bacterial loads: TMC 2.80±0.43 log cfu/g and TPC 2.52±0.28 log cfu/g)?

Author Response

Dear Reviewer 3,

We are grateful for the provided suggestions. Following Reviewer’s indications, we think that our manuscript has improved its scientific quality. All suggested revisions were performed.

In the following paragraphs we reported all changes.

Reviewer 3 suggestions: How do you explain that samples with higher salt content have higher TMC and TPC (The S4 samples showed a TMC average value of 2.86±0.44 log cfu/g and TPC 2.75±0.33 log cfu/g, respectively. S2 registered the following bacterial loads: TMC 2.80±0.43 log cfu/g and TPC 2.52±0.28 log cfu/g)?

Authors: As described in the Discussion section, these differences (TMC and TPC) between S4 and S2 products can be explained by the numerous manipulations strictly associated with the concept that marine salt can be considered a further contamination source, if the GHP were not correctly applied.

We confirm that neither the manuscript nor any parts of its content are currently under consideration or published in another journal.

All authors have approved the manuscript and agree with its submission to the FOODS Journal.

We appreciate the possibility to publish our paper and believe that our revised manuscript will be of interest to You and to the readers of Your journal.

Thank You for Your time and attention.

Best regards,

Gianluigi Ferri

Doctor in Veterinary Medicine (D.V.M.)

Ph.D. Student in Food Inspection

Faculty of Veterinary Medicine; University of Teramo, Italy.

Reviewer 4 Report

The authors gave a interesting and valuable work about revising salt content of codfish fillets during production chain. The microbiological indicators related to food safety for consumers were also investigated in this study. But there are some modifications needed. The review contents are as follows.

l Line 13: Alaska is not a country. So, ”(i.e., Norway, Denmark, Alaska, etc.)” should be deleted here. Please also pay attention to the font in this section.

l Lines 36-37: “70-85% and 2-3%” should be typed as “70%-85% and 2%-3%”. Same problem in Line 97.

l Lines 38-40: Due to the high total bacterial count? Or maybe caused by high water content or water activity directly. Please improve this section.

l Line 54: The “products’ shelf-life” should be typed as “shelf-life of products” or “shelf-life of packaging contents” OR “shelf-life of foodstuff”.  Same as line 65 “study’s aim” should be typed as “aim of study”.

l Lines 62-65: “It starts from raw materials: ....weight of 25kg.” should be moved to the “Materials and Methods “section. Besides, the fish sample weight in a “carton” should be given.

l Lines 89: Please add the reference information about “EU Reg. No. 1276/2011”, or give the necessary description.

l Line 90: “fish industries” should be typed as “fish producers”.

l Line 93: “2 kg salt” should be typed as “2 kg of NaCl”.

l Lines 94-95: Please check the accuracy of this statement. Besides, the “specimens” and “samples” should be unified throughout the text.

l Line 98: Please give the whole spelling of “aw” when firstly appear.

l Line 149: “activity water (aw)” or “water activity (aw)” ? 

l Table 3: The title of second column should be given as “Standards”. Besides, The classification of these cited standards should be given, for example, “11290-1:2017” should be typed as “ISO 11290-1:2017”. The reviewer suggested that the authors should add all the cited standards in the manuscript to the Reference section.

l Lines 198-207: The notes of Table4 should be deleted here. For this part was same with Table 1.

l Table 4 The salt contents and multilayer plastic film are the key factors in this study. The authors should give the differences description (with or without significantly differences) of pH, aw and NaCl between S4 and S2, as well as R4, R2, HPP OTR1-4, -2, HPP OTR2-4,-2. Besides, the detailed description for NaCl contents of every sample groups should be given in this section. This result is the key basis on which the authors give the possibility to reduce the use of salt during codfish fillets production chain.

l Figure 2: The unit of the vertical coordinate ( log cfu/g) should be added. Same problems in Figure 3, 4, and 5.

l Line 302, “in this research study”, the word “research” or “study” should be deleted one of them.

Author Response

Dear Reviewer 4,

We are grateful for the provided suggestions. Following Reviewer’s indications, we think that our manuscript has improved its scientific quality. All suggested revisions were performed.

In the following paragraphs we reported all changes.

Reviewer 4 suggestions:

Line 13: Alaska is not a country. So, ”(i.e., Norway, Denmark, Alaska, etc.)” should be deleted here. Please also pay attention to the font in this section.

Authors line 13: This sentence has been revised.

Lines 36-37: “70-85% and 2-3%” should be typed as “70%-85% and 2%-3%”. Same problem in Line 97.

Authors (lines38-39 and line 164): This sentence has been revised.

Lines 38-40: Due to the high total bacterial count? Or maybe caused by high water content or water activity directly. Please improve this section.

Authors (lines 42-44): This part has been implemented.

Line 54: The “products’ shelf-life” should be typed as “shelf-life of products” or “shelf-life of packaging contents” OR “shelf-life of foodstuff”.  Same as line 65 “study’s aim” should be typed as “aim of study”.

Authors (lines 88 and 96): This part has been modified.

Lines 62-65: “It starts from raw materials: ....weight of 25kg.” should be moved to the “Materials and Methods “section. Besides, the fish sample weight in a “carton” should be given.

Authors (lines 156-159): This sentence has been revised.

Lines 89: Please add the reference information about “EU Reg. No. 1276/2011”, or give the necessary description.

Authors (lines 150-152): A description has been introduced.

Line 90: “fish industries” should be typed as “fish producers”.

Authors (line153): This part has been modified.

Line 93: “2 kg salt” should be typed as “2 kg of NaCl”.

Authors (line 156): This part has been modified.

Lines 94-95: Please check the accuracy of this statement. Besides, the “specimens” and “samples” should be unified throughout the text.

Authors (lines 159-160): This part has been changed.

Line 98: Please give the whole spelling of “aw” when firstly appear.

Authors (line 110): This part has been changed.

Line 149: “activity water (aw)” or “water activity (aw)” ?

Authors (line 256): This part has been revised.

Table 3: The title of second column should be given as “Standards”. Besides, The classification of these cited standards should be given, for example, “11290-1:2017” should be typed as “ISO 11290-1:2017”. The reviewer suggested that the authors should add all the cited standards in the manuscript to the Reference section.

Authors (Table 3): Table has been modified.

Table 4 The salt contents and multilayer plastic film are the key factors in this study. The authors should give the differences description (with or without significantly differences) of pH, aw and NaCl between S4 and S2, as well as R4, R2, HPP OTR1-4, -2, HPP OTR2-4,-2. Besides, the detailed description for NaCl contents of every sample groups should be given in this section. This result is the key basis on which the authors give the possibility to reduce the use of salt during codfish fillets production chain.

Authors (lines 333-338): this part has been improved.

Figure 2: The unit of the vertical coordinate ( log cfu/g) should be added. Same problems in Figure 3, 4, and 5.

Authors: Figure 2, 3,4, and 5 have been modified.

Line 302 “in this research study”, the word “research” or “study” should be deleted one of them.

Authors (line 511): This part has been modified.

We confirm that neither the manuscript nor any parts of its content are currently under consideration or published in another journal.

All authors have approved the manuscript and agree with its submission to the FOODS Journal.

We appreciate the possibility to publish our paper and believe that our revised manuscript will be of interest to You and to the readers of Your journal.

Thank You for Your time and attention.

Best regards,

Gianluigi Ferri

Doctor in Veterinary Medicine (D.V.M.)

Ph.D. Student in Food Inspection

Faculty of Veterinary Medicine; University of Teramo, Italy.
